# OpenReview forum: "Efficient Reasoning with Hidden Thinking"
_ICLR.cc/2026/Conference — Submitted to ICLR 2026_

### Official Review · Reviewer_2DxQ · 2025-10-23

**Soundness:** 2
**Presentation:** 3
**Contribution:** 2
**Rating:** 2
**Confidence:** 4

**Summary:**

The paper considers multi-modal reasoning efficiency issues where input as multimodal (i.e., one image and one question), and textual reasoning with the final answers. Heima is proposed to condense each textual CoT steps into one specific token in the vocab via progressive distillation, and the paper additionally provide information-theoretic perspective analysis and tetxual CoT reconstruction via additional trained interpreter. The later part mainly is used to confirm the effectiveness of the proposed method, without other gains for the paper. The experimenal results validate the effective of Heima over two simple and weak baselines.

**Strengths:**

1. The targeted problem is interesting and valuable, but the aimed scope is limited.
2. The experiemntal results shows limited effectiveness.

**Weaknesses:**

1. the proposed method only consider one case that input is multi-modal, and the left part all based on textual reasoning and will not be affected by the input side. from this perspective, the contribution as first reasoning acceleration framework for MLLMs is weaken.
2. the progressive distillation is widely used in latent space reasoning, e.g., cocount.
3. despite the paper provide  information-theoretic perspective analysis, it seems no valueable insights, i.e., upper bound and lower bound, just extreme ideal situations.
4. also, the effectiveness can not be confirmed by the performance of interpreter, since there is no baseline or human evaluation. for example, although the reconsturcted sumamry by the interpreter is 4.1 out of 5 score (2.5/5 for caption), why 4 score is good enough? there is still a significant gap, weaken the statement of both  information-theoretic perspective analysis and value of the interpreter part.
5. other vicuna family model also is finetuned from llama family, it is more convincing to consider other family like qwen.
6. efficiency issue, since the method requires massive/multiple supervised fine-tuning in the progressive distillation stage.
7. baselines are too weak, whether or not the gain is worthy considering the cost to tune the model

**Questions:**

1. how many cot thinking tokens in total? what about the effect?

---

> ### Author Response · Authors · 2025-11-22
> **Author rebuttal to reviewer 2DxQ for weakness 1, weakness 2, and weakness 3**
>
> ### Rebuttal for Weakness 1
>
> Thanks for the comments. We went through the comments and believe that the reviewer agrees on our contribution as the first reasoning acceleration framework for MLLMs. But the reviewer thinks  that this contribution is not that significant since it explores tokens rather than vision in MLLMs.
>
> We first highlight that our contributions are not limited to   our compression framework for MLLM only. As mentioned in our introduction and contributions,  besides the compression framework, we ground the entire method in a formal information-theoretic analysis, and   introduce a novel interpreter framework for reconstructing the reasoning texts, which empirically verify the faithfulness of the compressed reasoning and greatly promote the  interpretability of thinking tokens.
>
> Furthermore, although we compress tokens, we indeed address the visual information in MLLMs.
> Heima explicitly considers the influence of multimodal inputs throughout the reasoning process, not only at the input stage. The interpreters are designed to reconstruct the reasoning trajectory in latent space, and the reconstructed reasoning clearly reflects visual information when there are only textual inputs, as shown in Figure 1 and 4.
> This demonstrates that the latent reasoning is indeed affected by and conditioned on the input image, rather than being purely textual. Therefore, our contribution as the first reasoning compression and acceleration framework for MLLMs remains valid and well-supported.
>
> ### Rebuttal for Weakness 2
>
> We believe this comment is based on a misunderstanding. The “progressive distillation” in Heima is fundamentally different from that in Coconut or other latent-reasoning methods.
>
> The compression methodologies of our method and Coconut are different. Specifically, we define new thinking token in vocabulary for each CoT stage, and these discrete thinking tokens are specifically generated during model inference. In contrast, Coconut directly utilizes the last hidden state (without converting to tokens) as the next input embedding.
> Furthermore, during our progressive distillation, both the thinking tokens and the rest textual CoT stages are used to compute the loss, different from Coconut, which masks out the questions and latent thoughts with only rest textual CoT stages for loss computations.
>
> Therefore, although both work adopt progressive distillation, the objective, representation, and supervision are entirely different.
>
> Progressive distillation is a general idea and we do not highlight progressive distillation as our main contributions. Compared with Coconut, besides the above mentioned difference,  our method introduces a novel interpreter framework for reconstructing varying-length CoTs from those compressed thinking tokens, which empirically verify the faithfulness of the compressed reasoning and greatly promote the interpretability of thinking tokens. Furthermore, our method provides an information-theoretic analysis quantifying the compression-induced information gap.
> This theoretical framework unifies our compression-based distillation method and provides the rigorous, intuitive guide for our Heima and interpreter design.
> The reconstruction a with interpreter and theoretic analysis are missing in previous works.
> We can see that our method is significantly different from Coconut with novel designs and analysis.
>
> ### Rebuttal for Weakness 3
>
> We thank the reviewer for this feedback. We believe there may be a **misunderstanding of our theory's purpose**. We respectfully clarify that our analysis is not intended to derive performance bounds (e.g., upper/lower bounds), but rather to serve as **the conceptual and guiding principle** for our entire framework.
>
> 1. Theory as Motivation, Not Post-Hoc Analysis: The reviewer seems to be looking for a post-hoc performance analysis, but our information-theoretic formulation is the ante-hoc motivation for our method. It formally models Heima as an information compression mechanism.
>
> 2. Guiding the Interpreter's Design: Our theory's primary contribution is formally defining the information gap ($I(Y;\mathrm{CoTs}\mid X,<\texttt{CoTs}>)$).
> The Heima interpreter is not an unrelated component; it was explicitly designed to empirically quantify this exact gap. This direct link between our theory and our architecture is one of the key novelty of our work.
>
> 3. Why Not Bounds? Deriving tight performance bounds would require significant, potentially unrealistic assumptions that would move away from our focus. Our goal is to provide the conceptual framework that justifies compression and guides the design of a system to validate it.

---

> ### Author Response · Authors · 2025-11-22
> **Author rebuttal to reviewer 2DxQ for weakness 4, weakness 5, and weakness 6**
>
> ### Rebuttal for Weakness 4
>
> We provide the interpreter evaluation to verify that the compressed latent reasoning retains sufficient information to reconstruct coherent and faithful reasoning traces.
>
> The scores we provided indicate the semantic consistency between the reconstructed and original CoTs, showing that the latent reasoning space encodes meaningful and verifiable reasoning patterns rather than random or collapsed representations.
>
> We show the prompt template for  GPT-4o in Appendix C when evaluating the similarity of the reconstructed tokens and the ground truth reasoning tokens.  We achieve a score of 4.1 for the summary stage, where score 4 corresponds to the level that  'Closely related with small differences' in Appendix 4, demonstrating high similarity. Our scores for other CoT stages  show that  the reconstructed CoT texts from the compressed thinking token   are  similar to the original texts.  It demonstrates that the reconstructed tokens closely align with the ground truth.
>
> If the reconstructed reasoning were unrelated to the original one, the scores would drop close to one. Therefore, the key point we want to deliver is not the absolute quality of the reconstruction itself, but the evidence that genuine reasoning indeed occurs within the latent space.
>
> ### Rebuttal for Weakness 5
>
> We provide the additional results with Qwen2.5-VL 7B in table below, and we believe the results verify the generalization of our method to the Qwen model family.
>
> | Model              | MMSar (# of Tokens) | MMBench (# of Tokens) | MMVet (# of Tokens) | MathVista (# of Tokens) | AI2D (# of Tokens) | Hallusion (# of Tokens) | Average |
> |---------------------|:------------------:|:---------------------:|:-------------------:|:-----------------------:|:------------------:|:-----------------------:|:--------:|
> | Qwen2.5-VL-7B       | 60.3 (51.6)        | 80.0 (10.2)           | 65.4 (137.7)        | 66.7 (202.9)            | 80.9 (2.6)         | 50.7 (69.0)             | **67.3** |
> | Qwen2.5-VL-7B CoT   | 65.2 (182.3)       | 82.1 (139.5)          | 69.4 (235.1)        | 67.8 (204.1)            | 84.4 (182.5)       | 64.8 (163.6)            | **72.3** |
> | Heima               | 61.1 (12.8)        | 81.9 (12.8)           | 59.5 (72.4)         | 58.7 (13.4)             | 79.3 (12.7)        | 63.1 (16.4)             | **67.3** |
>
> ### Rebuttal for Weakness 6
>
> The progressive distillation in Heima does not involve multiple full supervised fine-tuning runs. Instead, it is a lightweight, stage-wise distillation process with small LoRA adapters in LLM part and frozen visual encoders as mentioned in our experiment setups. The training cost with LoRA is relatively low.  It is a common practice to adopt LoRA to finetune  LLMs in prior works.
>
> Furthermore, although the progressive distillation needs to finetune the model for multiple rounds corresponding to the number of CoT stages, the number of rounds is not large
> (typically smaller than 5, e.g., LLaVA-CoT-100k only has 3 reasoning stages)
> and we only train one epoch in each round as shown in Appendix A. Thus, our method only needs to finetune the model on the dataset (not large with just 100K samples) for a few epochs (typically smaller than 5) with LoRA.  We believe that this training cost is common in most LLM related works.
>
>
> Moreover, this is a one-time training cost that enables substantial inference-time savings (token reduction and decoding latency), which aligns with the our goal of efficient reasoning. We will clarify this training-efficiency aspect in the revision.

---

> ### Author Response · Authors · 2025-11-22
> **Author rebuttal to reviewer 2DxQ for weakness 7 and question 1**
>
> ### Rebuttal for Weakness 7
>
> As mentioned in our introduction and  contribution,  our work is  the first reasoning acceleration framework for MLLMs that conducts reasoning in latent space by generating compact thinking tokens rather than verbose textual CoT. There are no other available token compression baselines for MLLMs.
> To demonstrate the compression performance, we compare with the original model without CoT enhancement, and the uncompressed model with verbose textual CoT finetuning. We believe that  it is a common practice to compare with the original uncompressed model, in the case that other compressed baselines are unavailable in this area.
>
> For the dataset selection, constructing a new multimodal reasoning dataset is extremely costly, as it requires large-scale stage-wise CoT annotations through API resources. In contrast, LLaVA-CoT-100K is a well-established and publicly available dataset, so we use it as a consistent and fair  dataset to evaluate our method.
>
> To ensure a fair comparison, we use the same base models and datasets during the comparison so that the only difference comes from our proposed reasoning compression. The comprehensive comparison demonstrates that  Heima achieves comparable accuracy with significantly fewer generated tokens (as little as 6\%), showing a clear efficiency gain.
>
> Furthermore, during our finetuning, we adopt LoRA as mentioned in our experiment setups. The training cost with LoRA is relatively low.  It is a common practice to adopt LoRA to finetune  LLMs in prior works.
> Note that it is a one-time training cost, and after training, it can generate much less tokens (as little as 6\%) for superior generation efficiency with greatly reduced inference cost.
> We believe that the improvement is well worth the cost. We will clarify this comparison strategy in the revision.
>
> ### Rebuttal for Question 1
>
> We adopt one thinking token for each CoT stage. In our experiments with LLaVA-CoT-100k, the dataset has three CoT stages (Summary, Caption, and Reasoning as shown in Figure 1) and thus we have three thinking tokens. As  mentioned in Line 178,  different data samples from the dataset share the same thinking token for the same CoT stage. So, we only use three thinking tokens for the whole dataset.  In Table 1 and 2, the number of generated tokens also include the final conclusion tokens besides thinking tokens. Thus the number of generated tokens are typically larger than 3.
>
> We adopt one thinking token for each CoT stage, and we show the ablation study for number of thinking tokens in Figure 5.
> From the left of Figure 5, we provide the ablation results with different number of thinking tokens, and the results show that the single thinking token for each CoT achieves the best results.
> Meanwhile, as shown in the middle of Figure 5, we also consider adopt the dynamic number of thinking tokens for the compression. We adopt the number of thinking tokens according to the specific retention ratios of the original number of tokens for the CoT. The results with different retention ratios show the unstable and varying accuracy performance, and the single thinking token shows better performance than all the results achieved with retention ratios.
> Note that, we show the number of generated tokens on MMStar at the right side of Figure 5, and we identify that, when the retention ratio becomes 0.7, then the number of generated tokens are more than the original textual CoTs, which shows the dynamic number of thinking tokens are not suitable compared to the fixed number of thinking tokens.

---

> > ### Comment · Reviewer_2DxQ · 2025-11-25
> >
> > Thank you for your clarification, I decide to raise my inital score.

---

> > > ### Author Response · Authors · 2025-11-25
> > >
> > > We sincerely thank reviewer 2DxQ, and we truly appreciate the constructive feedback and the recognition of our work.

---

### Official Review · Reviewer_S84H · 2025-10-28

**Soundness:** 3
**Presentation:** 3
**Contribution:** 2
**Rating:** 6
**Confidence:** 4

**Summary:**

This paper addresses the inefficiency of CoT reasoning in MLLMs, which generates lengthy textual tokens and incurs high computational costs. To tackle the problem, this paper proposes Heima, a framework that compresses verbose CoTs into compact thinking tokens in latent space through progressive distillation. In addition, an interpreter is designed to reconstruct reasoning texts from thinking tokens for validation. Experiments show that Heima maintains or even surpasses the accuracy of the original CoT model while using only about 6% of the tokens, demonstrating efficient and effective latent-space reasoning.

**Strengths:**

1. The motivation of this work is interesting, as compressing Chain-of-Thought (CoT) to accelerate reasoning can effectively reduce computational costs, particularly in resource-intensive MLLM scenarios.
2. The proposed method is straightforward and effective, with comprehensive experiments validating its efficacy.
3. The information-theoretic analysis provides a solid theoretical foundation for the effectiveness of CoT compression, which is highly appreciated.

**Weaknesses:**

1. Although the idea of compressing CoT in MLLMs is promising, similar approaches have been extensively explored in the context of LLMs, suggesting that the methodological novelty may be somewhat limited.
2. Given that CoT reasoning demonstrates significant performance gains on more challenging tasks, the experimental evaluation should be extended to include more complex reasoning benchmarks, such as MathVision or OlympiadBench.
3. Beyond the LLaVA-Next and Llama3.2-11B-Vision models, it remains unclear whether the method generalizes effectively to other architectures, such as the Qwen-VL series. Theoretically, this should be a universally applicable approach.

**Questions:**

1. Can this method be effectively applied to long-chain CoT scenarios, such as those in QvQ or Virgo? If so, would the compression ratio be higher compared to standard CoT, and would there be a significant degradation in performance?
2. Additional experimental results should be provided to further substantiate the claims and explore the method's boundaries.

---

> ### Author Response · Authors · 2025-11-22
> **Author rebuttal to reviewer S84H for weakness 1, weakness 2, and weakness 3**
>
> We thank the reviewer for this thoughtful comment.
>
> ### Rebuttal for Weakness 1
>
> We highlight that our method is significantly different from previous works focusing on reasoning efficiency for LLMs. Compared with prior works on text-only LLMs, our work is the first to: (1) explore latent reasoning for multimodal LLMs; (2) ground the entire method in a formal information-theoretic analysis, which guides our distillation method and motivates    the interpreter design; (3)  introduce a novel interpreter framework for reconstructing the reasoning texts, which empirically verify the faithfulness of the compressed reasoning and greatly promote the  interpretability of thinking tokens.
>
>
> ### Rebuttal for Weakness 2
>
> Thanks for the suggestion.
> We agree that evaluating on more challenging reasoning benchmarks, such as MathVision and OlympiadBench, would further strengthen the paper.
>
> Our current experiments focus on LLaVA-CoT-100K, which already provides a diverse set of multimodal reasoning tasks and allows us to carefully validate the core mechanisms of reasoning compression and interpreter-based verification under controlled settings.
>
> Extending to new datasets such as MathVision and OlympiadBench would require additional efforts. (i) The datasets need to be processed first before training.
> Specifically,  the datasets should  provide not only the correct answers, but also the reasoning process.  Furthermore, it is better to construct a structured reasoning process  with multiple consecutive  reasonable steps for more effective thinking distillation, rather than one single long reasoning step. (ii) Moreover,  we need to fine-tune a new model on the new dataset before the progressive distillation, which is computationally expensive.
> We plan to include more evaluations in our follow-up works.
>
>
> ### Rebuttal for Weakness 3
>
> We provide the additional results with Qwen2.5-VL 7B in table below, and we believe the results verify the generalization of our method to the Qwen model family.
>
> | Model              | MMSar (# of Tokens) | MMBench (# of Tokens) | MMVet (# of Tokens) | MathVista (# of Tokens) | AI2D (# of Tokens) | Hallusion (# of Tokens) | Average |
> |---------------------|:------------------:|:---------------------:|:-------------------:|:-----------------------:|:------------------:|:-----------------------:|:--------:|
> | Qwen2.5-VL-7B       | 60.3 (51.6)        | 80.0 (10.2)           | 65.4 (137.7)        | 66.7 (202.9)            | 80.9 (2.6)         | 50.7 (69.0)             | **67.3** |
> | Qwen2.5-VL-7B CoT   | 65.2 (182.3)       | 82.1 (139.5)          | 69.4 (235.1)        | 67.8 (204.1)            | 84.4 (182.5)       | 64.8 (163.6)            | **72.3** |
> | Heima               | 61.1 (12.8)        | 81.9 (12.8)           | 59.5 (72.4)         | 58.7 (13.4)             | 79.3 (12.7)        | 63.1 (16.4)             | **67.3** |

---

> ### Author Response · Authors · 2025-11-22
> **Author rebuttal to reviewer S84H for question 1 and question 2**
>
> ### Rebuttal for Question 1
>
> **Extending to New Datasets.**
>
> It is feasible to apply Heima to long-chain reasoning scenarios such as QvQ or Virgo. Our framework can naturally adapt to phase-wise multi-step reasoning distillation. But it requires additional efforts to process the data and finetune the model on new datasets.
>
> Specifically, as shown in our methodology, our frame supports reasoning under multiple stages, without specific constraints on the number or contents of stages. Thus, for long-chain CoT scenarios, we can construct datasets that decompose the long complex reasoning progress into several successive and simpler CoTs across multiple stages.  Each stage forms a self-contained reasoning segment. Then we can apply  our progressive distillation to compress each stage into  a single stage-specific thinking token. Our method naturally supports multi-stage reasoning compression without specific requirements on  the number or contents of reasoning steps.
> The LLaVA-CoT-100k dataset in our experiments is an example which splits the reasoning with multiple simpler steps including question summary, image caption, reasoning, and conclusion.
>
> To actual extend Heima to long-chain reasoning scenarios, additional efforts are required such as dataset processing for multi-step reasoning, and model fine-tuning on new datasets.
>
> With our designed interpreters that visualize and verify the reasoning process in latent space, we believe this stage-wise formulation enables Heima to effectively handle long and complex reasoning tasks while preserving interpretability.
>
> Considering that such experiments require building new long, multi-stage reasoning datasets with stage-wise CoT annotations  and model fine-tuning on new datasets, we may not be able to provide the corresponding experimental results within the rebuttal period.
>
> **Compression Ratio.**
>
> The compression ratio of new datasets depends on how to split  COT into multiple stages and the length of each stage.
> Since our method does not have specific requirements on the number or contents of multiple COT stages.
> (i) For new datasets,  if the data are split into less stages with less thinking tokens (one thinking token corresponds to one COT stage), a higher compression ratio can be achieved compared with more stages with more thinking tokens.
> (ii) Within each stage, if it contains more texts,  a higher compression ratio can be achieved since more textual tokens within each stage are compressed into one single think token.
>
> Since typical COT stages are verbose with multiple textual tokens,  our method with one single thinking token to replace all of them can lead to significant compression ratios. Our experimental results demonstrate that the accuracy on  LLaVA-CoT-100K degrades marginally, and our  interpreters successfully reconstructs the thinking tokens back into
> textual reasoning traces, proving our effectiveness. We believe that the accuracy degradation is marginally on new datasets.
>
> ### Rebuttal for Question 2
>
> We provide additional experimental results with Qwen2.5-VL 7B in rebuttal for weakness 3, which further support our claims and demonstrate the generalizability of Heima. We hope this addresses the reviewer’s concern.

---

> ### Comment · Reviewer_S84H · 2025-11-27
>
> After reviewing the rebuttal, two significant concerns remain unresolved:
>
> 1. The improvement achieved by Heima on Qwen2.5-VL-7B is marginal, suggesting that Heima may lack generalizability within a universal MLLM framework.
>
> 2. It appears that Heima requires fine-tuning to achieve optimal performance on different downstream tasks. I thought that Heima is a general token compression framework that could be readily adapted to various tasks. If training is necessary for adaptation to downstream tasks, the applicability of Heima would be substantially limited.

---

> > ### Author Response · Authors · 2025-11-27
> > **Follow up answers to Reviewer S84H**
> >
> > We thank the reviewer for the follow-up comments and would like to clarify two points.
> >
> > 1. **The Qwen2.5-VL-7B results are not marginal.**
> >
> > We highlight that our compression work focuses on reducing reasoning tokens with improved efficiency. As shown in the Qwen results and other model results in Table 1 and 2, the number of generated tokens are significantly reduced. For example, compared with Qwen2.5-VL-7B without CoT enhancement, Heima uses up to 15$\times$ fewer tokens. Compared with CoT-enhanced models which use much more tokens for reasoning, the efficiency improvement of Heima becomes more significant,  such as  6\% token number for LLaVA-Next-Vicuna-7B CoT (Table 2) or Qwen2.5-VL-7B CoT on  MathVista.
> >
> > Meanwhile, with improved efficiency, our accuracy are similar to uncompressed models. Compared with the original non-CoT models, our method can typically lead to higher or same accuracy,  such as our 67.3 v.s. 67.3 from Qwen2.5-VL-7B, or  our 58.0 v.s. 52.1 from Llama3.2-11B Vision.  Compared with CoT-enhanced models,  our method can achieve competitive accuracy (such as our 53.8 v.s. 55 from LLaVA-Next-Vicuna-7B CoT) with much less tokens.
> >
> > In summary, we believe that the Qwen results are consistent with our results in the main paper, which demonstrates that Heima leads to significant efficiency improvements with much fewer tokens, while achieving better or competitive accuracy. We believe that efficiency improvements are not  marginal.
> >
> >
> > 2. **Heima does not require task-specific finetuning.**
> >
> > We believe that this is a misunderstanding, and our method does not need task-specific finetuning.  As discussed in Section 4.1, we finetune the models on the LLaVA-CoT-100k dataset. During  accuracy evaluation, we use multiple challenging zero-shot benchmarks to verify the effectiveness of our proposed method, including MMStar, MMBench V1.1, MMVet, MathVista, AI2D, and HallusionBench.  The models are finetuned on  LLaVA-CoT-100k with LoRA only, and never finetuned on downstream tasks such as MMStar or MMVet.
> > When applying Heima to Qwen2.5-VL-7B, we only train the lightweight LoRA module with data from LLaVA-CoT-100k. After the compression is finished, Heima can be readily adapted to various downstream tasks with the accuracy reported in the Qwen results table.
> > We can see that the training is not necessary for adaptation to downstream tasks.
> >
> > For more complex downstream tasks such as MathVision or OlympiadBench, it is better to apply our framework on more complex CoT datasets with math-style reasoning progress. Once the compression progress is finished, the model can be applied for evaluation on downstream tasks with complex reasoning progress such as MathVision or OlympiadBench, without further additional fine-tuning. Currently, in our work, we target the same reasoning benchmark as LLaVA-CoT-100k [1]  for the verification of the effectiveness for our proposed framework.  LLaVA-CoT-100k  is not a math specific CoT dataset, and thus we do not evaluate on math-focused downstream benchmarks with complex math reasoning steps.
> >
> > Sorry for any misunderstanding. In our rebuttal, when we discuss new datasets, we discuss the steps required if we need to compress the reasoning CoTs in new datasets, where new datasets are regarded as the finetuning data, rather than downstream evaluation data.
> > If we would like to test on new downstream evaluation benchmarks with complex mathematical reasoning progress, additional task-specific finetuning is not necessary. But to achieve a better performance, it is better to finetune the model with related reasoning data, such as enhancing the CoT capability of Qwen model by finetuning on CoT dataset LLaVA-CoT-100k.
> >
> > ---
> >
> > [1] LLaVA-CoT: Let Vision Language Models Reason Step-by-Step

---

> ### Comment · Reviewer_S84H · 2025-11-28
>
> Thanks for your clarification. However, given the overall quality of the paper, I decide to keep my original score.

---

### Official Review · Reviewer_Dkrj · 2025-10-31

**Soundness:** 3
**Presentation:** 3
**Contribution:** 2
**Rating:** 6
**Confidence:** 3

**Summary:**

This paper proposes Heima, a framework for compressing Chain-of-Thought (CoT). Heima compresses the content of a CoT into an implicit Tinking Token, significantly reducing the number of tokens, and designs an interpreter to restore the CoT Token to text. The paper trains LlaVA-CoT and compares it with the original model on multiple datasets, demonstrating that Heima achieves excellent results in saving tokens.

**Strengths:**

1. This paper proposes a framework for compressing CoT, shrinking CoT into a single token, significantly reducing the length of the model's response while maintaining its quality.

2. This paper uses information-theoretic analysis and an interpreter to verify the effectiveness and rationality of the compression. The interpreter successfully reconstructs the reasoning process, confirming the minimization of the information gap.

**Weaknesses:**

1. The experimental setup in this paper primarily focuses on short-step reasoning. For longer, more complex reasoning tasks, such as intricate mathematical proofs or logical deductions, compressing the reasoning content into a single token raises questions about its impact on key information retention and model capability, requiring further experimental analysis. Adding longer-step reasoning might improve the paper's generalization ability.

2. In the experiments of Section 4, the paper only compares Heima with the pedestal model. Perhaps adding comparisons with other methods to improve reasoning efficiency would better highlight Heima's high efficiency.

3. In CoT task reasoning, the model may need to dynamically adjust subsequent reasoning paths based on intermediate results, such as in Rethinking. Heima compresses reasoning information into a single token. Will this affect the aforementioned dynamic adjustment, and will it limit the model's exploratory reasoning capabilities?

**Questions:**

Please see the  weaknesses.

---

> ### Author Response · Authors · 2025-11-22
> **Author rebuttal to reviewer Dkrj**
>
> Thank you for the constructive and insightful suggestions.
>
> ### Rebuttal for Weakness 1
>
> For long and complex reasoning, our framework can naturally adapt to phase-wise multi-step reasoning progress. As shown in our methodology, our frame supports reasoning under multiple stages, without specific constraints on the number or contents of stages.
>
> (i) For long and complex reasoning progress, we can construct datasets that decompose the long complex reasoning progress into several successive and simpler CoTs across multiple stages.  Each stage forms a self-contained reasoning segment. Then, we can apply  our progressive distillation to compress each stage into  a single stage-specific thinking token. Our method naturally supports multi-stage reasoning compression without specific requirements on  the number or contents of reasoning steps.
>
> (ii) We experiment with the LLaVA-CoT-100k dataset, which already  splits the original long reasoning progress into multiple stages, including question summary, image caption, reasoning, and conclusion steps. Our method demonstrates superior performance for this dataset.
>
> (iii) For longer, more complex reasoning tasks, such as intricate mathematical proofs or logical deductions, we can still split the reasoning into multiple stages and compress each stage with  one single  thinking token,  instead of  compressing the whole reasoning of multiple stages with one token.
>
> (iv) With our designed interpreters that visualize and verify the reasoning process in latent space, we believe this stage-wise formulation enables Heima to effectively handle long and complex reasoning tasks while preserving interpretability.
>
> Considering that such experiments require building new long, multi-stage reasoning datasets with stage-wise CoT annotations through external APIs like LLaVA-CoT [1], we may not be able to provide the corresponding experimental results within the rebuttal period due to limited API resources.
>
>
> ### Rebuttal for Weakness 2
>
> To the best of our knowledge, our work is  the first reasoning acceleration framework for MLLMs that conducts
> reasoning in latent space by generating compact thinking tokens rather than verbose textual CoTs.  It is hard to find other baselines for direct comparison in the area of compressing MLLM Reasoning.
>
> Some previous works  such as Coconut [2] and CoCoMix [3] focus on reasoning efficiency for text-only large language models. They use last latent states as the next input embedding or concept mixing within a pure language setting.
>
> In contrast, Heima is designed for multimodal LLMs and additionally provides interpreters to reconstruct the compressed CoTs, enabling explicit verification of the latent reasoning process. We highlight that our interpreter can successfully  reconstruct the visual information from compressed token inputs  using a LLM without any image inputs as shown in Figure 1, demonstrating that the thinking token is performing reasoning with visual information. This unique design greatly promote the  interpretability of our work,  which is absent in previous text-only  works.
> As there are fundamental differences in both modality and objective, direct comparison with those text-only methods is not applicable.
>
> ### Rebuttal for Weakness 3
>
> We appreciate this insightful question.
> Although Heima compresses the explicit CoT text into compact thinking tokens, our analysis through the corresponding interpreters shows that the model continues to perform the reasoning in latent space rather than memorizing fixed outcomes.
>
> Specifically, the interpreters reconstruct intermediate reasoning traces from different thinking tokens, and these reconstructed sequences clearly exhibit context-dependent adjustment and exploration, similar to explicit CoT reasoning. This demonstrates that the latent reasoning process still evolves adaptively based on intermediate states, even after compression.
>
> Therefore, while the reasoning CoTs are compressed into thinking tokens, the latent reasoning dynamics remain across different stages and are still adaptive, allowing the model to maintain exploratory reasoning capabilities comparable to standard CoT reasoning.
>
>
> -----
>
> [1] LLaVA-CoT, a visual language model capable of spontaneous, systematic reasoning
>
> [2] Training Large Language Models to Reason in a
> Continuous Latent Space
>
> [3] LLM Pretraining with Continuous Concepts

---

### Official Review · Reviewer_Y7SG · 2025-10-31

**Soundness:** 3
**Presentation:** 3
**Contribution:** 2
**Rating:** 4
**Confidence:** 4

**Summary:**

The paper proposes **Heima**, a latent CoT–compression framework for MLLMs that replaces verbose textual chains of thought with a small number of **special “thinking tokens”**, learned via **progressive distillation**. It also introduces **LLM interpreters** that map the hidden states of these tokens back to text to estimate the “information gap,” alongside a brief information-theoretic argument (via DPI) that compressed tokens should retain task-relevant info.

**Strengths:**

- Experiments on MMStar, MMBench, MMVet, MathVista, AI2D and HallusionBench claim comparable accuracy to CoT baselines while generating far fewer tokens.
- **Interpreter idea for auditing latent reasoning**: training text-only LLMs to reconstruct reasoning from hidden states is a useful diagnostic to probe whether compressed tokens preserve semantics.

- **Token-efficiency results**: On several benchmarks, the method reports sizable **token reductions** (e.g., to ~6–10% of CoT tokens in places) with modest accuracy loss and sometimes gains relative to LLaVA-CoT.

**Weaknesses:**

- **Insufficient novelty and unclear justification.** The method is highly similar to *Coconut* (latent reasoning through continuous hidden states) and other latent CoT approaches such as *Cocomix*. The paper’s claim of being the first to extend such latent reasoning to MLLMs feels incremental.
- **Lack of rationale for multimodal adaptation.** The authors do not explain why a Coconut-style framework should work particularly well in the multimodal setting. Simply applying a latent CoT designed for text-only reasoning to MLLMs requires more theoretical or empirical justification.
- **Incomplete efficiency analysis.** The paper emphasizes token reduction but lacks concrete wall-clock latency, memory, or FLOP statistics under matched decoding setups.

**Questions:**

- Apart from the application domain, what are the most substantial differences between Heima and Coconut? Could other latent CoT approaches (e.g., Cocomix, CODI) also be applied in MLLMs?
- Coconut’s results were not particularly strong in pure NLP reasoning tasks. Why does a similar staged-distillation framework yield competitive results here?
- Why did the authors choose the Coconut paradigm over other latent reasoning methods? What multimodal properties make it more suitable?
- How does Heima perform when handling inputs with heavier visual content, such as multi-image or video scenarios?
- The experiments are limited to a narrow set of base models. How would **Heima** perform on more recent and widely adopted architectures such as **Qwen2.5-VL** or **InternVL 3**, which have stronger multimodal grounding?

---

> ### Author Response · Authors · 2025-11-22
> **Author rebuttal to reviewer Y7SG for weakness 1, weakness 2, and weakness 3**
>
> ### Rebuttal for Weakness 1
>
> Thank you for the comments. We believe there may be some misunderstandings regarding our method, which is **totally different from Coconut** in both formulation and implementation. We kindly encourage the reviewer to revisit the technical details in our paper, and we provide clarifications below.
>
> We respectfully disagree with the characterization that our work is merely an incremental extension of Coconut. There are several substantive differences:
>
> (1) Scope and modality: Coconut studies latent reasoning in pure text LLMs, whereas our method Heima focuses on multimodal LLMs to address the challenges of reasoning with both visual and textual inputs.
>
> (2)  Methodology difference: The compression methodologies of our method and Coconut are different. Specifically, we define new thinking token in vocabulary for each CoT stage, and these discrete thinking tokens are specifically generated during model inference. In contrast, Coconut directly utilizes the last hidden state (without converting to tokens) as the next input embedding.
> Furthermore, during our progressive distillation, both the thinking tokens and the rest textual CoT stages are used to compute the loss, different from Coconut, which masks out the questions and latent thoughts with only rest textual CoT stages for loss computations.
>
>
> (3)  Interpretability demonstration: Our method introduces a novel interpreter framework for reconstructing varying-length CoTs from those compressed thinking tokens, which empirically verify the faithfulness of the compressed reasoning and greatly promote the
> interpretability of thinking tokens. This is significantly different from previous works such as Coconut which do not provide such mechanisms to check or interpret the compressed token or the latent reasoning progress with solid demonstration.
>
> (4) Theoretical contribution: Coconut focuses on the latent reasoning paradigm and its empirical behavior in LLMs, but it does not present any theoretical contribution.
> In contrast, our Heima framework provides an information-theoretic analysis quantifying the compression-induced information gap.
> Our specially designed interpreters and experiments on multimodal zero-shot benchmarks then demonstrate that compression preserves reasoning capability while significantly improving efficiency.
> This theoretical framework unifies our compression-based distillation method and provides the rigorous, intuitive guide for our Heima and interpreter design.
>
>
> ### Rebuttal for Weakness 2
>
> We think the reviewer’s comment is based on an incorrect premise, as our method is not Coconut-style (as we explained the method difference for Weakness 1), and we did not such transplant text-only latent reasoning into MLLMs.
>
> We did not adopt the last hidden state as the embedding input for the next token prediction during reasoning progress.
> In our method, we compress the CoTs into discrete thinking tokens through step-wise distillation, We further design the separate interpreters to reconstruct those compressed thinking tokens for the interpretability.
>
> In contrast, Coconut does not address (1) multimodal latent reasoning, nor (2) verifiable reasoning compression (i.e., whether reasoning in latent space is faithful, interpretable, and reconstructable)
>
> ### Rebuttal for Weakness 3
>
> We emphasize that, in comparing our method with other baselines, the prefill phase remains identical since the input is the same. The key difference lies in the reasoning progress during the decoding phase, where latency, memory usage, and FLOPs scale directly with the number of generated tokens.
> Thus, reducing the number of generated tokens during decoding can significantly accelerate the reasoning progress and reduce the latency with improved throughput.
> This is why we mainly report the counts of generated tokens as the core efficiency metric in our results.
> To further demonstrate this, we list profiling results for the latency and generated tokens on A6000 GPU in the following table.
> We identify significant latency reduction when the number of generated tokens are reduced.
> Based on the following table and Table 1 in our paper, the LLaVA-CoT baseline for the MMBench dataset requires about 154 tokens, which takes more than 1.6 seconds according to the profiling results. In contrast, our method generates fewer than 16 tokens and completes in less than 0.2 seconds, which is over 8$\times$ faster.
> We agree that adding wall-clock latency, memory, and FLOPs can further strengthen the evaluation, and we will include them in the revision.
>
> | Number of Generated Tokens | Total Latency (s) |
> |:---------:|:--------------:|
> | 16        | 0.20         |
> | 32        | 0.41         |
> | 128       | 1.64         |
> | 256       | 3.30         |
> | 512       | 6.62         |

---

> ### Author Response · Authors · 2025-11-22
> **Author rebuttal to reviewer Y7SG for Question 1, Question 2, and Question 3**
>
> ### Rebuttal for Question 1
>
> The key differences are not limited to the application domain. Our Heima differs fundamentally from Coconut and other latent CoT works in several aspects:
>
> (1) Representation: Heima adopts discrete thinking tokens (i.e., special tokens registered in vocabulary rather than hidden states), not continuous hidden states as next embedding input in Coconut or mixed latent embeddings in Cocomix.
>
> (2) Training Objective: We perform step-by-step distillation for CoT compression in different stages, instead of next-state latent prediction or ungrounded latent modeling.
>
> (3) Verifiability: Heima introduces a reconstruction-based interpreter to verify that compressed reasoning is faithful and decodable, which is absent in Coconut, Cocomix, and CODI.
>
> (4) Information Preservation Guarantee: We provide a formal information-theoretic analysis of reasoning retention, which prior latent CoT methods do not formulate.
> This serves as the guiding principle for our entire framework, modeling Heima as an information compression mechanism and motivating the core design of the Heima interpreter, which is built to empirically quantify this exact information gap.
>
> ### Rebuttal for Question 2
>
> This question is based on an incorrect premise, as Heima is not a “Coconut-style staged distillation” method. We have discussed the detailed difference in the above replies.
>
> Furthermore, as discussed in Line 323-340, we design an interpreter to investigate the information gap highlighted in Theorem 3.1 and Remark 3.3.  By comparing the reconstructed reasoning sentences with the original textual CoTs,  it demonstrates that the reconstructed reasoning closely aligns in semantics with the original CoTs. Thus, the compression induced information gap is regarded as minimal, thereby confirming that reasoning with thinking tokens preserve the essential reasoning capability.
> Different from previous works, our novel interpreter design and theoretic discussion present that our thinking tokens preserve the essential reasoning capability, leading to the outstanding compression performance.
>
> ### Rebuttal for Question 3
>
> We believe that this question is based on a false premise. We did not choose the Coconut paradigm, nor do we adopt its core design (last hidden state as embedding input for next token prediction).
>
> Our design is motivated by a straight-forward idea to reduce the number of reasoning tokens for reasoning efficiency. Next, based on our theoretic analysis,  the  interpreter is proposed to reconstruct the compressed thinking with better transparency and interpretability. The  interpreter confirms that reasoning with thinking tokens preserve the essential reasoning capability, demonstrating our effectiveness. Similar design is not available in those latent-only methods like Coconut.
>
> We do not choose a paradigm. Our tried multiple  configurations with many experiments, such as the case without progressive or recovery training shown in Table 1 and ablation study.  Our solid experiments verify our effectiveness.
> Our design choice is determined by our initial idea,  insightful theoretic analysis,  comprehensive framework construction, and  solid experiment verification.
> Simply choosing a paradigm is not able to lead to our work.
>
> In summary, our design choices are not inherited from Coconut, but are motivated by multimodal reasoning fidelity and verifiability, which Coconut does not address.

---

> ### Author Response · Authors · 2025-11-22
> **Author rebuttal to reviewer Y7SG for Question 4 and Question 5**
>
> ### Rebuttal for Question 4
>
> Heima can  work with visual inputs of various types or length,   such as multiple images or videos. As shown in our methodology, the reasoning compression focuses on the reasoning tokens,  rather than the visual encoder part.  Our method does not change the visual input part  in $X$ of Equation (3), and thus can support  visual inputs of various types or lengths.
>
> In the case with longer visual inputs  (e.g., multi-image or video), we believe that our method can still perform well.
> Longer inputs may lead to longer reasoning since it needs to obtain and process more information from more visual inputs. Our method can naturally support reasoning with multiple reasoning stages since we do not have any constraints on the number or the contents of reasoning stages. Thus, if the long reasoning texts from long visual inputs can be well split into multiple reasonable CoT stages, our method can be applied directly with outstanding performance. The current results on LLaVA-CoT-100k, which split the reasoning into question summary, image caption, reason, and conclusion stages, demonstrate our superior performance under multi-stage reasoning compression.
>
> ### Rebuttal for Question 5
>
> The main contribution of our method is compressing reasoning CoTs into thinking tokens and reconstructing them with an interpreter, which works on top of any MLLM decoder.
> Therefore, we believe our method can benefit from models with stronger visual grounding capabilities.
>
> In our experiments, we start from a model fine-tuned on LLaVA-CoT-100K first, and then adopt the progressive distillation with our method.
> In the following table, we apply our method for the Qwen2.5-VL-7B model with first finetuning (on LLaVA-CoT-100K) and then progressive distillation. The results verify the effectiveness of our proposed method on Qwen-VL model family.
>
> Extending our Heima training pipeline to other larger models from Qwen2.5-VL or InternVL3  requires  much more computing resources  and training hours. We will include these experiments in the revision if available.
>
> | Model              | MMSar (# of Tokens) | MMBench (# of Tokens) | MMVet (# of Tokens) | MathVista (# of Tokens) | AI2D (# of Tokens) | Hallusion (# of Tokens) | Average |
> |---------------------|:------------------:|:---------------------:|:-------------------:|:-----------------------:|:------------------:|:-----------------------:|:--------:|
> | Qwen2.5-VL-7B       | 60.3 (51.6)        | 80.0 (10.2)           | 65.4 (137.7)        | 66.7 (202.9)            | 80.9 (2.6)         | 50.7 (69.0)             | **67.3** |
> | Qwen2.5-VL-7B CoT   | 65.2 (182.3)       | 82.1 (139.5)          | 69.4 (235.1)        | 67.8 (204.1)            | 84.4 (182.5)       | 64.8 (163.6)            | **72.3** |
> | Heima               | 61.1 (12.8)        | 81.9 (12.8)           | 59.5 (72.4)         | 58.7 (13.4)             | 79.3 (12.7)        | 63.1 (16.4)             | **67.3** |

---

> ### Author Response · Authors · 2025-11-26
>
> Dear Reviewer Y7SG,
>
> We sincerely thank you again for the valuable feedback and constructive questions.
> We would like to kindly draw your attention to our rebuttal and newly added results, which illustrates clear difference from Coconut and directly address your concerns:
>
> 1. We have included latency analysis and profiling results.
>
> 2. We also added additional experimental results on the Qwen2.5-VL-7B model.
>
> 3. All weaknesses and questions you raised have been carefully addressed in our rebuttal.
>
> We hope these updates clarify the novelty, multimodal applicability, and efficiency of Heima, and we would greatly appreciate it if you could kindly revisit our responses.
>
> Thank you again for your time and thoughtful review.
>
> Warm regards,
>
> Authors

---

> > ### Comment · Reviewer_Y7SG · 2025-11-27
> >
> > Thank you for your response. It has addressed my concerns, and I have updated my score accordingly.

---

### Author Response · Authors · 2025-12-03

We sincerely thank the Area Chair and all reviewers for their detailed feedback and constructive evaluations of our paper.
We really appreciate the active engagement of reviewers during rebuttal and the subsequent score improvements (**[4, 6, 6, 2] → [6, 6, 6, 6]**), which reflect a shared recognition of our contributions  after  detailed sincere  discussions.

In response to the valuable suggestions, we have provided comprehensive explanations, new latency profiling, and extended experiments (e.g., Qwen2.5-VL-7B generalization). The key improvements are summarized below:

**Clear distinction from Coconut/Cocomix and other latent CoT methods**: We clarified the fundamental methodological differences, including (1) discrete thinking tokens instead of hidden-state recursion, (2) step-wise distillation with supervised CoT stages, (3) multimodal support, and (4) a new interpreter mechanism absent in prior works.

**New experiments validating generalization to Qwen2.5-VL-7B**: We added a full evaluation for another model  Qwen2.5-VL-7B under the same LLaVA-CoT-100K training pipeline, demonstrating that Heima generalizes well to stronger visual backbones and larger model families.

**Concrete latency profiling showing >8$\times$ decoding acceleration**: We provided matched decoding-time measurements on an A6000 GPU, showing that reducing reasoning tokens directly yields real wall-clock gains, with Heima requiring <16 tokens and <0.2s decoding versus around 154 tokens with >1.6s for LLaVA-CoT.

Besides, we further clarify our applicability to multi-image or video reasoning, our theoretical contribution with its connection to the interpreter, training cost, and latent-space reasoning fidelity.

These additional analyses with expanded experiments and stronger clarifications have significantly strengthened the paper’s novelty, empirical validity, and overall contribution.

---

### Meta-Review · Area_Chair_7UPi · 2026-01-13

**Summary:**

The paper proposes Heima, a framework for compressing Chain-of-Thought (CoT) reasoning in Multimodal Large Language Models (MLLMs). It replaces textual reasoning steps with compact "thinking tokens" in the latent space via progressive distillation. The authors also introduce an "interpreter" to decode these latent tokens back into text to verify reasoning fidelity and provide an information-theoretic analysis of the compression. The reviews for the paper were borderline. The main concerns were: (1) novelty of the paper, (2) weak baselines, (3) no particularly insight in the theory and (4) fairly limited experimental setup.

After reading the reviews and rebuttal, I recommend rejection in the current form. Despite the rebuttal, I think the novelty of the paper is fairly limited and empirical comparison is very weak (with no strong baselines and ablations). The addition of a decoder to is a useful feature but it is not sufficient for acceptance of the paper.

**Reviewer Concerns:**

Reviewer Y7SG and Reviewer 2DxQ correctly identified that the core mechanism and progressive distillation is not particularly novel. The shift from text-only LLMs to MLLMs, while noteworthy, does not particularly highlight the novelty of the paper. The authors justification of novelty is fairly minor and it is important to properly ablate the differences to showcase the benefit.

Reviewer 2DxQ noted the "baselines are too weak," making it difficult to understand the true gain of Heima over SoTA efficiency methods. I agree with this assessment even after the rebuttal. Without proper baselines, it is hard to assess the empirical impact of the work.

**Reviewer Scores:**

I think Reviewer Y7SG and Reviewer 2DxQ concerns are not fully addressed in my opinion. This paper remains borderline according to my assessment.

---

### Decision · Program_Chairs · 2026-01-26

Reject